# A Bayesian psychophysics model of sense of agency

Roberto Legaspi [1,2] & Taro Toyoizumi[1,2]

Sense of agency (SoA) refers to the experience or belief that one's own actions caused an external event. Here we present a model of SoA in the framework of optimal Bayesian cue integration with mutually involved principles, namely reliability of action and outcome sensory signals, their consistency with the causation of the outcome by the action, and the prior belief in causation. We used our Bayesian model to explain the intentional binding effect, which is regarded as a reliable indicator of SoA. Our model explains temporal binding in both self-intended and unintentional actions, suggesting that intentionality is not strictly necessary given high confidence in the action causing the outcome. Our Bayesian model also explains that if the sensory cues are reliable, SoA can emerge even for unintended actions. Our formal model therefore posits a precision-dependent causal agency.

[1] Laboratory for Neural Computation and Adaptation, RIKEN Center for Brain Science, Wako City, Japan. [2] RIKEN CBS-OMRON Collaboration Center, 2-1 Hirosawa, Wako City, Saitama 351-0198, Japan. Correspondence and requests for materials should be addressed to R.L. (email: roberto.legaspi@riken.jp) or to T.T. (email: taro.toyoizumi@riken.jp)

  1

Sense of agency (SoA) is the registration[1] that the self initiates actions to influence its external environment[2]. It therefore accompanies voluntary actions[3–6], allows oneself to feel distinct from others[7–9], and be responsible for its own actions[2,6,10,11]. Studies show SoA emerges from, and is particularly sensitive to any disruption in, the congruous flow of intentional actions to expected sensory outcomes[12]. Crucially, the degradation of this experience characterizes certain psychiatric and neurological disorders[13–15]. For example, studies show schizophrenic patients tend to attribute someone else's actions to themselves. Despite its significance[16–18], the literature still lacks the computational principles that can elucidate SoA.

We theorize SoA as the confidence in one's perception of the action-outcome effect, and that it is consistent (e.g., spatially or temporally) with the hypothesis that the action caused the outcome. We adapted the model of Sato et al.[19] that was originally used to explain the ventriloquism effect as a Bayesian estimate of a common source behind the consistency of the audiovisual stimuli, akin to being the common cause[20] of the audiovisual integration. Formalizing SoA by this Bayesian psychophysics principle distinguishes our theory from existing works.

We compared the predictions of our model with the results of two pertinent intentional binding studies. Intentional binding, which is the perceived compression of the time interval between voluntary action and its outcome, has been reported as a reliable implicit measure of SoA and has been used in a large number of studies providing valuable analyses on the temporal perception of action-outcome effects and the nature of SoA[21]. The seminal experiment of Haggard et al.[3] investigated the perceived action-outcome timing effects in three conditions: voluntary wherein the subject intentionally presses a button, involuntary wherein muscle twitches of the subject's hand are induced by a transcranial magnetic stimulation (TMS) applied to the motor cortex, and sham TMS wherein the TMS on the parietal cortex produces audible clicks but no movement (hereafter, voluntary, involuntary, and sham conditions, respectively). Haggard and colleagues computed the time interval between the perceived action timings (with the timings of either voluntary actions, muscle twitches, or audible TMS clicks as control experiment) and the perceived timings of subsequent tone stimuli. They showed that voluntary actions produced intentional binding, involuntary muscle twitches produced repulsion, i.e., prolonged opposite perception of the action-outcome intervals, and audible TMS clicks produced neither binding nor repulsion. Hence, they posit intentionality is necessary to achieve action-outcome binding.

The second pertains to the study of Wolpe et al.[22], which investigated the contribution of cue integration to intentional binding by manipulating the reliability of the consequent tone relative to a background white noise. Such manipulation resulted in three levels of tone uncertainty conditions, namely low, intermediate, and high uncertainty. Their analyses showed that when tone reliability was reduced, the perceptual shift in tone timing towards the action was increased.

Although Bayesian integration was proposed as a general principle behind SoA[14,23], it was unknown whether the observed action-outcome temporal compression and repulsion effects are consistent with Bayesian principles, and if indeed the case, the question is how. Our Bayesian model reproduces the above empirical results on intentional binding based on a computational principle. Further, it goes beyond timing estimations by exposing the underlying Bayesian mechanisms that possibly drove the temporal binding. Our Bayesian model explains the perceived compressed action-outcome time interval is more consistent with the prior belief of the causal role of one's action in producing the immediate outcome and thus increases the confidence in the Bayesian estimate assuming the causal case, modeled as SoA.

Moreover, our model explains intentional binding as a specific class of the more general notion of causal binding. Our Bayesian model predicts that intentional binding generally happens on a per-trial basis, yielding a bimodal distribution of the perceived action-outcome interval. Lastly, the model also predicts that if the sensory input signals are perceived as reliable (precise), SoA may arise even for unintended actions, which serves as a testable theory for future SoA experiments.

## Results

**Bayesian inference model of action-outcome temporal binding.** We considered the experimental setup of intentional binding where a subject presses a button (i.e., the action) and a tone (i.e., the outcome) sounds 250 ms after the button press. The true action and outcome timings are thus described by $t_A^* = 0$ ms and $t_O^* = 250$ ms, respectively, but they are unknown to the subject. The task for the subject is to accurately report her perceived timings of the button press and tone. We assume the arrival of relevant sensory input informing the timing of each of these physical events involves sensory delay $d$ and jitter of variance $\sigma^2$ due to sensory noise. Thus, the arrival time $\tau_A$ of sensory input that signals the action timing is assumed to be generated from a Gaussian distribution, $\mathcal{N}(t_A^* + d_A, \sigma_A^2)$, with mean $t_A^* + d_A$ and variance $\sigma_A^2$. Similarly, the arrival time $\tau_O$ of sensory input that signals the outcome timing is generated from $\mathcal{N}(t_O^* + d_O, \sigma_O^2)$.

The brain often resolves such ambiguity in sensory inputs by integrating multiple sensory cues akin to the Bayesian "ideal observer"[24]. Hence, we model a Bayesian observer who estimates action timing $t_A$ and outcome timing $t_O$ based on the corresponding noisy sensory inputs arriving at time $\tau_A$ for the action and $\tau_O$ for the outcome. The conditional probability distributions of $\tau_A$ and $\tau_O$ that the Bayesian observer uses are modeled as Gaussian distributions

$$
\begin{aligned}
P(\tau_A | t_A) &\propto \exp\left(-\frac{(\tau_A - t_A)^2}{2\sigma_A^2}\right) \\
P(\tau_O | t_O) &\propto \exp\left(-\frac{(\tau_O - t_O)^2}{2\sigma_O^2}\right),
\end{aligned}
\tag{1}
$$

with mean $t_A$ and $t_O$, and variance $\sigma_A^2$ and $\sigma_O^2$ for action and outcome, respectively. It is noteworthy that sensory delays $d_A$ and $d_O$ are not included in Eq. (1) for the reason we describe in the next paragraph.

Before studying the binding effect, let us consider simple baseline conditions. In one baseline condition, the action timing is reported by the subject without the presentation of an outcome tone. If no prior knowledge is available, the Bayesian observer reports the action timing that maximizes the conditional probability distribution in Eq. (1). Hence, the estimated action timing $\hat{t}_A = \tau_A$ is solely determined by the noisy sensory input informing the action timing. In this case, the model predicts that the distribution of $\hat{t}_A$ is $\mathcal{N}(t_A^* + d_A, \sigma_A^2)$. The mean and SD of $\hat{t}_A$ in the baseline condition were experimentally reported, e.g., Haggard's results in the voluntary condition suggest $d_A = 6$ ms and $\sigma_A = 66$ ms (refer to Table 1 in Methods for all condition-based $d_A$ and $\sigma_A$ values). Importantly, we assume that the observer does not take into account sensory delay $d_A$ in Eq. (1). If the Bayesian observer included its effect, it could compensate for this delay and report unbiased timing, which was not the case in the experiment. Therefore, we assume that the observer was unable to take into account the sensory delay in Eq. (1). In the other baseline condition, the subject passively listens to a tone and reports its timing. This case goes parallel to the above case and the model predicts that the estimated tone timing is $\hat{t}_O = \tau_O$,

**Table 1 List of Bayesian model parameters and their values**

| Baseline parameters | | | | | Operant parameters | |
|---|---|---|---|---|---|---|
| All are temporal values in unit ms | $d_A$ | $\sigma_A$ | $d_O$ | $\sigma_O$ | $\mu_{AO} = 230$ ms　$\sigma_{AO} = 10$ ms | $P(\xi = 1)$ |
| ● Set A: Reported by Haggard et al.[3] | | | | | ● Set C: Obtained by our Bayesian model | |
| Voluntary action | 6 | 66 | | | Voluntary action | 0.9 |
| Involuntary action (TMS-induced muscle twitch) | 83 | 83 | | | Involuntary action (TMS-induced muscle twitch) | 0.9 |
| Sham TMS (audible click only) | 32 | 78 | | | Sham TMS (audible click only) | 0.1 |
| Auditory tone | | | 15 | 72 | | |
| ● Set B: Reported by Wolpe et al.[22] | | | | | ● Set D: Obtained by our Bayesian model | |
| Voluntary action | −8 | 75 | | | | |
| Low tone uncertainty | | | 35 | 61 | Low tone uncertainty | 0.9 |
| Intermediate tone uncertainty | | | 46 | 66 | Intermediate tone uncertainty | 0.6 |
| High tone uncertainty | | | 95 | 90 | High tone uncertainty | 0.5 |

which is distributed according to $\mathcal{N}\left(t_O^* + d_O, \sigma_O^2\right)$. The comparison of this model prediction to Haggard's experiment, e.g., would be $d_O = 15$ ms and $\sigma_O = 72$ ms (refer to Table 1).

Next we study the effect of binding when the subject makes an action and then listens to the outcome tone, commonly referred to as the operant condition. In this case, the Bayesian observer makes an inference not only based on the conditional probability distribution in Eq. (1) but also based on the prior distribution of $t_A$ and $t_O$. Adapting the Bayesian model of the ventriloquism effect[19], we assume the prior distribution depends on the observer's belief whether the action caused the outcome, i.e., the causal case: $\xi = 1$, or the action and the outcome are unrelated, i.e., the acausal case: $\xi = 0$:

$$P(t_A, t_O | \xi) \propto \begin{cases} \exp\left(-\frac{(t_O - t_A - \mu_{AO})^2}{2\sigma_{AO}^2}\right), & (\xi = 1) \\ 1. & (\xi = 0) \end{cases} \quad (2)$$

The action causes the outcome in the causal case ($\xi = 1$) so that the outcome timing involves a typical delay $\mu_{AO}$ with respect to the action timing and a Gaussian-distributed jitter of SD $\sigma_{AO}$. The outcome is caused by something other than the action in the acausal case ($\xi = 0$) so that $t_A$ and $t_O$ are independent. Lastly, we define $P(\xi)$ as the prior for each belief: $P(\xi = 1)$ for the causal case and $P(\xi = 0) = 1 - P(\xi = 1)$ for the acausal case. We hypothesize the estimation of $\xi$ to be essential for the perception of causality and SoA (explained below).

Given a pair of sensory inputs at $\tau_A$ and $\tau_O$, the Bayesian observer estimates the most probable timing for the action and the outcome, and whether these observations are consistent with the causal case. According to the Bayesian estimation theorem, the maximum-a-posteriori (MAP) estimate $(\hat{t}_A, \hat{t}_O, \hat{\xi})$ of the corresponding pair of physical sensory timing $(t_A, t_O)$ and the causal variable $\xi$ is given by

$$\left(\hat{t}_A, \hat{t}_O, \hat{\xi}\right) = \arg \max_{t_A, t_O, \xi} P(t_A, t_O, \xi | \tau_A, \tau_O), \quad (3)$$

where $P(t_A, t_O, \xi | \tau_A, \tau_O)$ is the posterior probability distribution of $(t_A, t_O, \xi)$ given the sensory inputs $(\tau_A, \tau_O)$. Hence, whether the Bayesian observer estimates the action-outcome effect to be causal or not depends on the posterior ratio comparing the causal case ($\xi = 1$) and the acausal case ($\xi = 0$), namely

$$r \equiv \max_{t_A, t_O} P(t_A, t_O, \xi = 1 | \tau_A, \tau_O) / \max_{t_A, t_O} P(t_A, t_O, \xi = 0 | \tau_A, \tau_O). \quad (4)$$

Causality is detected if the confidence in the causal estimate is greater than that in the acausal case, i.e., $r > 1$. The MAP estimate

of Eq. (3) is then given by (see Methods for the derivation)

$$\left(\hat{t}_A, \hat{t}_O, \hat{\xi}\right) = \begin{cases} \left(\tau_A + \frac{\sigma_A^2}{\sigma_{tot}^2}(\tau_O - \tau_A - \mu_{AO}), \tau_O - \frac{\sigma_O^2}{\sigma_{tot}^2}(\tau_O - \tau_A - \mu_{AO}), 1\right), & (r > 1) \\ (\tau_A, \tau_O, 0), & (r < 1) \end{cases} \quad (5)$$

with $\sigma_{tot}^2 \equiv \sigma_A^2 + \sigma_O^2 + \sigma_{AO}^2$. This indicates, on one hand, that perceptual shift does not happen if the causality is not detected ($\hat{\xi} = 0$)—the time estimates for action and outcome simply reflect the corresponding sensory signals in this case. On the other hand, perceptual shift happens if the causality is detected ($\hat{\xi} = 1$)—the action and outcome timing attract each other in the form of binding if $\tau_O - \tau_A > \mu_{AO}$ and repel each other in the form of repulsion if $\tau_O - \tau_A < \mu_{AO}$. The magnitude of perceptual shift for the action and outcome timing depends on coefficients $\sigma_A^2/\sigma_{tot}^2$ and $\sigma_O^2/\sigma_{tot}^2$, respectively, implying that perceptual shift is greater for a more unreliable stimulus. This model predicts that the occurrence of binding, repulsion, or no perceptual shift is trial-dependent, influenced by the noisy sensory signal $\tau_O - \tau_A$ informing the action-outcome interval. We denote the probability of detecting causality (i.e., $\hat{\xi} = 1$) by $P_c$ (see Methods for its analytical expression). $P_c$ increases with larger $P(\xi = 1)$ and smaller $\sigma_{AO}$ if $\sigma_{AO} \ll \sigma_A, \sigma_O$.

**Proposed measure of SoA.** Separate from the judgement of causality described above, we also directly quantify the confidence in the causal MAP estimate

$$CCE = \max_{t_A, t_O} P(t_A, t_O, \xi = 1 | \tau_A, \tau_O) \quad (6)$$

and we postulate this quantity to be a possible indication of the pre-reflective feeling of agency (FoA; see Discussion). The analytical expression of confidence in causal estimate (CCE) in Methods yields the following requirements to have high CCE: (i) the timing of sensory signals must be consistent with the causation of the outcome by the action, namely $\tau_O - \tau_A \approx \mu_{AO}$; (ii) the causal prior probability $P(\xi = 1)$ must be high; (iii) the sensory inputs must be precise, i.e., the amplitudes $\sigma_A$ and $\sigma_O$ of sensory jitter must be small enough. Furthermore, by computing for the peak of the conditional probability distribution, instead of integrating over $t_A$ and $t_O$, CCE does not only indicate the causation of the outcome by the action but is also sensitive to the accuracy of the action and outcome timing estimates. We therefore posit SoA as encapsulation and manifestation of several pertinent aspects, which include temporal consistency in the action-outcome effect, the prior belief of an action causing the outcome, and the reliability of the perceived sensory signals. Hence, our Bayesian model coherently explains not just SoA that arises from the causation of the outcome by the action but also one that

is influenced by the reliability of the different agency cues—a precision-dependent causal agency.

**Simulation results and model predictions.** Here we briefly describe how we obtained the parameter values used in our simulation (but see Methods for more details about the model fitting and simulation). Fitting of $d_A$, $d_O$, $\sigma_A$, and $\sigma_O$ is straightforward; they are suggested by the means and SDs of the reported subjects' baseline estimation errors (Table 1-Sets A and B in Methods). After fixing these parameters, the model is left with three free parameters, $\mu_{AO}$, $\sigma_{AO}$, and $P(\xi = 1)$. As described in Eq. (5), $\mu_{AO}$ has an important role in determining whether binding or repulsion happens in each experimental condition. A fixed value of $\mu_{AO} = 230$ ms successfully accounts for this qualitative behavior in all the six experimental conditions (three from Haggard et al.[3] and three from Wolpe et al.[22]) that we study. The analytical expressions in Methods suggest that $\sigma_{AO}$ and $P(\xi = 1)$ have a largely overlapping role in detecting causality. Causality is more likely detected if $\sigma_{AO}$ is small or $P(\xi = 1)$ is large, although the exact mechanisms are slightly different. At least one of these two parameters needs to be adjusted according to the conditions to account for the experimental observations. For simplicity, we fix $\sigma_{AO} = 10$ ms to be a small enough constant to permit noticeable perceptual shift and adjust $P(\xi = 1)$ (see Table 1 for the parameter values in six experimental conditions) to account for two observations in each condition, namely the perceptual shifts in the action timing and the outcome timing.

Our results show that our simple Bayesian model qualitatively reproduces the perceptual shifts that were reported in the study by Haggard et al.[3] (Fig. 1). Consistent with their findings, our Bayesian observer inferred the perceived action and outcome timings to shift towards each other in the voluntary condition, resulting in compressed temporal intervals between the action and outcome perceptual shifts. However, reversed and prolonged perceptual shifts were observed in the involuntary condition. The model also reproduced no appreciable perceptual shifts in the sham condition.

Our Bayesian model predicts binding and repulsion to increase with stronger causal prior (Fig. 2). From Eq. (5), the amount of binding or repulsion is given by $(\tau_O - \tau_A) - (\hat{t}_O - \hat{t}_A)$, which is

$(\tau_O - \tau_A - \mu_{AO})(\sigma_A^2 + \sigma_O^2)/\sigma_{tot}^2$ in the causal case ($\hat{\xi} = 1$) and none otherwise ($\hat{\xi} = 0$). As the sensory signals are distributed according to $\tau_A \sim \mathcal{N}(t_A^* + d_A, \sigma_A^2)$ and $\tau_O \sim \mathcal{N}(t_O^* + d_O, \sigma_O^2)$, the average of $\tau_O - \tau_A - \mu_{AO}$ factor is $m = t_O^* - t_A^* + d_O - d_A - \mu_{AO}$. Hence, the sign of $m$ determines whether binding or repulsion is predicted on average. With the current set of parameters, $m$ is positive in the voluntary condition, yielding binding, and negative in the involuntary condition, yielding repulsion (schematically drawn in Fig. 3a). Perceptual shift is almost zero regardless of the causal prior $P(\xi = 1)$ in the sham condition, because $m \approx 0$. We chose $P(\xi = 1) = 0.1$ for this under-constrained sham condition, assuming that causality would not be frequently detected.

Our Bayesian model provides interesting insights on what possibly drives the perceived action-outcome temporal compression and repulsion effects. We empirically observed sensory delay $d$ to increase with larger SD $\sigma$ of the Gaussian-distributed jitter (observed in both Haggard et al.[3] and Wolpe et al.[22]; see Table 1 in Methods). This may imply that, as action or outcome ambiguity is increased due to noise (greater $\sigma$) for increased sensory uncertainty, more time would be needed (greater $d$) for a sensory input to reach the subject's perceptual threshold for temporal awareness in the baseline condition. Thus, because of $m$'s dependency on $d_O - d_A$, binding more likely happens when the outcome is unreliable (i.e., with large $d_O$) and repulsion more likely happens when the action is unreliable (i.e., with large $d_A$).

To further illustrate the model prediction from our simulations, we plotted separately the action and outcome perceptual shifts for the three conditions as functions of the temporal disparity $\tau_O - \tau_A$ (c.f. Eq. (5)). Indeed, our data show that for instances in which $\tau_O - \tau_A > \mu_{AO}$, action awareness is delayed (positive action shift, Fig. 3b) and outcome tone is anticipated (negative outcome shift, Fig. 3c), thereby demonstrating binding. The opposite happens when $\tau_O - \tau_A < \mu_{AO}$, thereby demonstrating repulsion in both action and outcome awareness (Fig. 3b, c, respectively). We then plotted how the model's MAP estimates on the action-outcome interval are affected by the sensory time difference $\tau_O - \tau_A$ in the baseline (here, $\hat{\xi} = 0$ is forced; Fig. 3d) and operant (Fig. 3e) conditions. We observe from the baseline condition that the MAP estimates follow sensory inputs,

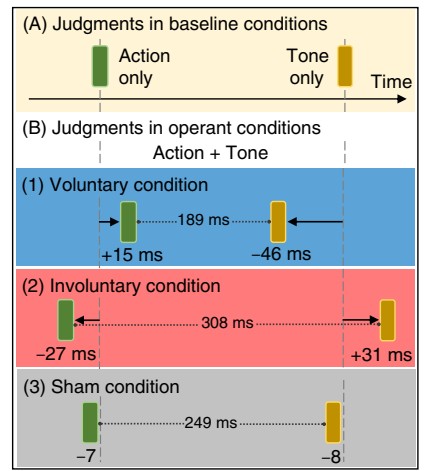

Results from Haggard et al. (2002)

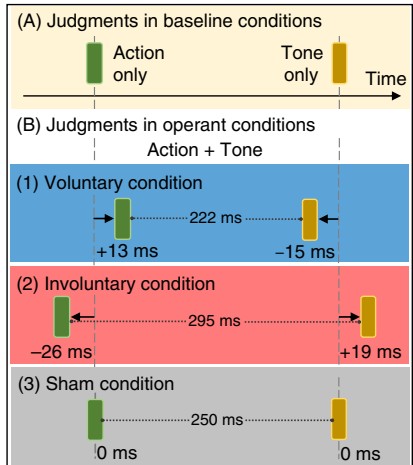

Simulation results with Bayesian model

**Fig. 1** Qualitative replication of the empirical results reported by Haggard et al.[3]. Each subject's mean judgment error in the single-event baseline condition was subtracted from the mean judgment error for the corresponding event in the operant condition. This resulted in the values underneath the blocks that indicate the magnitude and direction to which the temporal perceptions shifted. A positive perceptual shift informs delayed awareness and a negative shift informs anticipated awareness. The action and outcome timings are perceived to shift towards each other in the voluntary condition. In contrast, they are perceived to repulse in the involuntary condition. There is no discernible perceptual shift in the sham condition

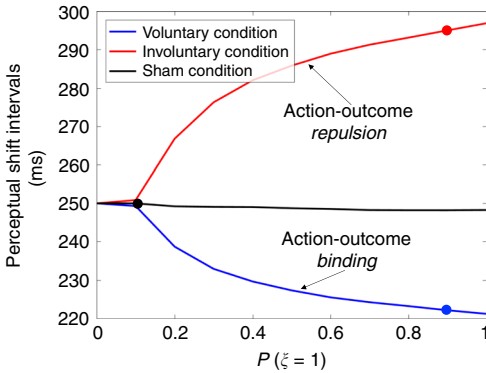

**Fig. 2** Bayesian model predictions of the influence of causal prior strength on action-outcome perceptual shifts. The best estimates of the Bayesian model (in Fig. 1) were obtained from different causal priors, specifically, $P(\xi = 1)$ is 0.9, 0.9, and 0.1 (marked by the colored dots) for the voluntary, involuntary, and sham conditions, respectively. The intervals between the action and outcome perceptual shifts shrink in the voluntary, but widen in the involuntary, condition with a strong causal prior. Minimal changes in perceptual shifts are predicted for the sham condition even with a strong causal prior

$\hat{t}_O - \hat{t}_A = \tau_O - \tau_A$, whereas the perception of action and outcome timings shifted towards the prior mean, $\hat{t}_O - \hat{t}_A \approx \mu_{AO}$, in the voluntary and involuntary conditions but not so much in the sham condition with weak causal prior. Therefore, our model is agnostic as to whether the action is self-intended or unintended. Binding towards $\hat{t}_O - \hat{t}_A \approx \mu_{AO}$ will happen, be it in the opposite direction, as long as the action is believed to have caused the outcome. This suggests that causality is the phenomenon that underlies intentional binding, and likely SoA, with self-intended causality being a specific case. The temporal window of $\tau_O - \tau_A$ for detecting causality is wider in the voluntary and involuntary conditions than in the sham condition.

We then examined how the prior belief in causation affects our proposed measure for SoA in Haggard's experimental setup. Our model predicts CCE to strengthen together with the causal prior but its strength differs depending on the conditions even at the same strength of the prior (Fig. 4a). Interestingly, $d_A$ and $\sigma_A$ are the only parameters of our Bayesian model that differentiate the three conditions in this figure. As we described above, these two parameters are empirically correlated such that the delay $d_A$ increases with larger $\sigma_A$. Hence, the difference in CCE in the three conditions can be attributed to the inequalities in SDs of the subjects' action timing estimation errors in the three conditions: $\sigma_A^{Vol} < \sigma_A^{Sham} < \sigma_A^{Invol}$ as per the data of Haggard et al.[3] Haggard et al. speculated that the unexpected and surprising quality of the TMS-induced movement could account for the repulsion effect in the involuntary condition. We suggest that this surprise might have introduced uncertainty in the perception of action input signals. Hence, although subjects were certain of the nature of their voluntary actions, they could be less certain of the proprioception signals induced by TMS, which could explain the inequalities in $\sigma_A$. As a result, the model gives $CCE^{Vol} > CCE^{Sham} > CCE^{Invol}$ according to requirement (C), i.e., reliable sensory inputs, for having high CCE when compared at the same strength of the causal prior.

The relation between CCE and SoA becomes clear when we analyze them with the fitted values of the causal prior ($P(\xi = 1) = 0.9$ for the voluntary and involuntary conditions and $P(\xi = 1) = 0.1$ for the sham condition as indicated in Table 1). Figure 4b plots CCE on a per-trial basis as functions of the temporal

disparity $\tau_O - \tau_A$ (c.f. the analytical expression for CCE in Methods). CCE in the voluntary condition has a higher peak than the involuntary condition as we described above (due to small $\sigma_A$ in the voluntary condition for the requirement (C)). In both voluntary and involuntary conditions, CCE diminishes as $\tau_O - \tau_A$ moves farther from $\mu_{AO}$ because of the requirement (A) of small $|\tau_O - \tau_A - \mu_{AO}|$ for having high CCE. Finally, CCE for the sham condition takes much lower values than the voluntary or involuntary conditions because of the requirement (B) of large $P(\xi = 1)$ for having high CCE.

In a similar fashion, we then examined the underlying psychophysical mechanisms that could account for the temporal binding observed by Wolpe et al.[22], in which three uncertainty levels (high, intermediate, and low uncertainty) of the outcome stimulus were tested. We use the Bayesian model that was used to reproduce the Haggard's experiments with the same values of $\mu_{AO}$ and $\sigma_{AO}$ but adjusted the strength of the causal prior $P(\xi = 1)$ to fit the reported action timing and outcome timing in each condition. We used $P(\xi = 1) = 0.9$, 0.6, and 0.5 for low, intermediate, and high tone uncertainty conditions, respectively (see Table 1 and Methods). This means that the prior belief in causation decreases with the tone uncertainty, which is plausible. (Alternatively, we could increase $\sigma_{AO}$, which produces similar results; see above discussion on model fitting.)

Our model reproduces the experiments of Wolpe et al.[22] (Fig. 5a), qualitatively explaining the temporal binding they observed in terms of a single, coherent cue integration formulation. The Bayesian estimate of the action-outcome intervals shift towards $\hat{t}_O - \hat{t}_A \approx \mu_{AO}$, as per the causal temporal prior in Eq. (2) when causality is detected. On the one hand, the magnitude of the shift is greater when the outcome uncertainty is high (c.f. Eq. (5)). However, on the other hand, causality is less frequently detected when the outcome uncertainty is high with the reduced causal prior. These two opposing effects are summarized in Fig. 5b. The model can qualitatively reproduce the experiments if the former effect is more dominant. Quantitatively, however, the latter effect is necessary to mitigate the former effect.

Next, we plot how the Bayesian estimate of the action-outcome interval, $\hat{t}_O - \hat{t}_A$, depends on the sensory inputs $\tau_O - \tau_A$. The perceived intervals faithfully follow the sensory inputs in the baseline condition (Fig. 5c), where all trials are acausal ($\hat{\xi} = 0$) by definition. In the operant condition (Fig. 5d), the Bayesian estimate shifts towards the prior assumption $\hat{t}_O - \hat{t}_A \approx \mu_{AO}$ when the sensory inputs are highly consistent with the prior $\tau_O - \tau_A \approx \mu_{AO}$ and, thus, when the causality is detected ($\hat{\xi} = 1$). Otherwise, the estimate of action-outcome intervals follows sensory inputs. The temporal window of $\tau_O - \tau_A$ for detecting causality is wider when the outcome uncertainty is lower.

Next, we quantify again CCE as a possible measure of SoA. CCE diminishes with outcome uncertainty even when compared at the same level of causal prior (Fig. 6a). Hence, CCE explicitly depends on the outcome uncertainty. When plotted as functions of temporal disparity, with the specific causal priors obtained for each outcome uncertainty condition, the peak values of CCE noticeably differ across the uncertainty conditions (Fig. 6b). This is because of the different values of the outcome uncertainty $\sigma_O$ but also partly because of the different values of the causal prior. In all conditions, CCE falls off with the disparity of sensory inputs from the prior mean, $|\tau_O - \tau_A - \mu_{AO}|$. This fall-off is milder when the uncertainty is lower. These results clearly manifest again three basic requirements of CCE as follows: (i) the consistency of sensory inputs with the causal prior; (b) strong prior belief in causality; and (c) reliable sensory inputs.

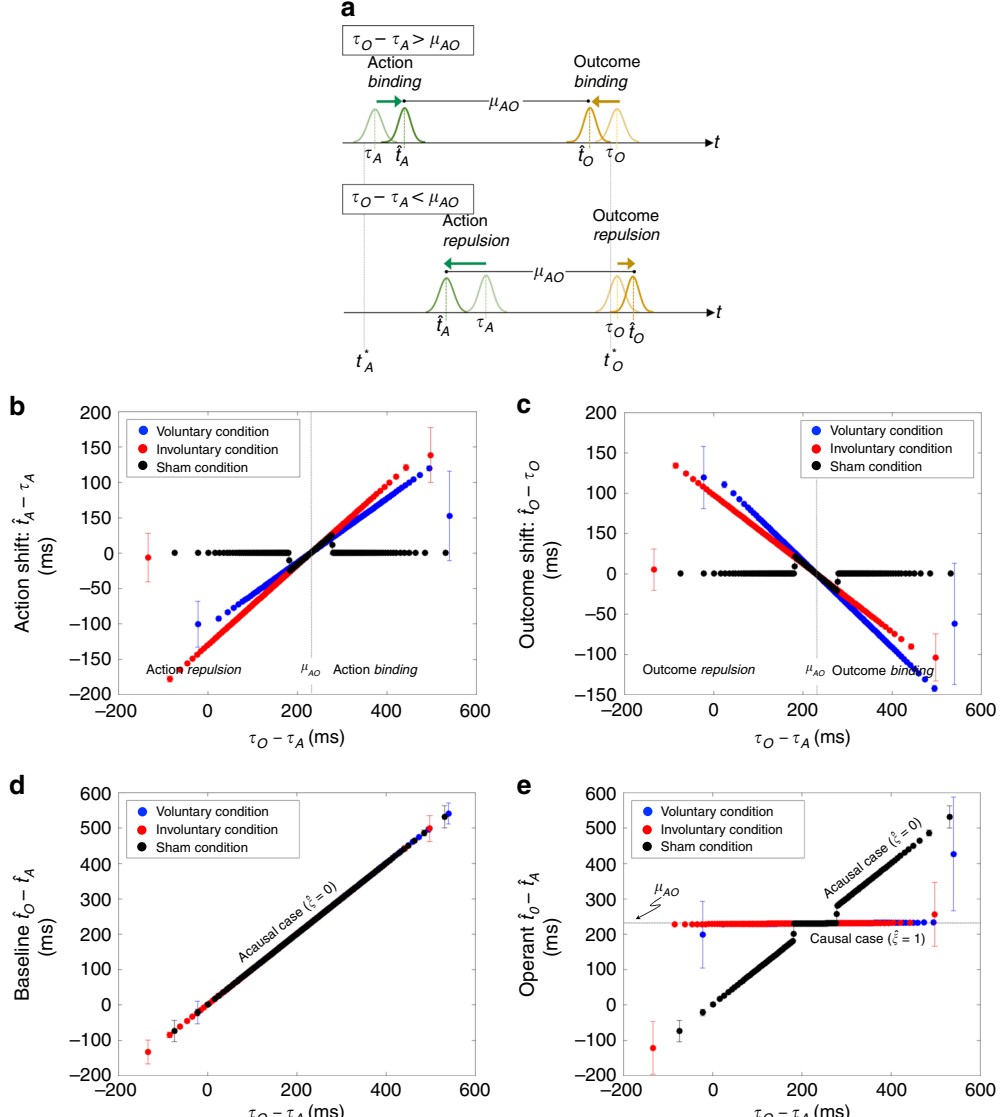

**Fig. 3** Bayesian model predictions of trial-to-trial perceptual shifts and timing intervals. **a** Our Bayesian model predicts that (shown schematically) if $\tau_O - \tau_A > \mu_{AO}$ action and outcome binding will happen. Otherwise, i.e., $\tau_O - \tau_A < \mu_{AO}$, action-outcome repulsion will occur. In both cases, the perceived timings in the baseline move (compress or stretch) towards the temporal consistency $\hat{t}_O - \hat{t}_A \approx \mu_{AO}$ in the operant condition. **b**, **c** When $\tau_O - \tau_A > \mu_{AO}$, there is positive perceptual shift in action awareness ($\hat{t}_A - \tau_A > 0$) and negative perceptual shift in outcome awareness ($\hat{t}_O - \tau_O < 0$). The opposite happens when $\tau_O - \tau_A < \mu_{AO}$. Both binding and repulsion occur in both voluntary and involuntary conditions, but very little effect in the sham condition. **d** The Bayesian estimates follow the sensory inputs in the baseline condition, i.e., $\tau_O - \tau_A \approx \hat{t}_O - \hat{t}_A$, where all trials are acausal ($\hat{\xi} = 0$) by definition. **e** The Bayesian estimate shifts towards the prior assumption, $\hat{t}_O - \hat{t}_A \approx \mu_{AO}$, when the sensory inputs are highly consistent with the prior, $\tau_O - \tau_A \approx \mu_{AO}$, and therefore when causality is detected ($\hat{\xi} = 1$). Otherwise, the estimate of action and outcome timings follow the sensory inputs. The fitted causal prior $P(\xi = 1)$ is 0.9, 0.9, and 0.1 for the voluntary, involuntary, and sham conditions, respectively (as in Fig. 2). The per-trial results are grouped accordingly into bins of width 200 (randomly chosen), and the mean and SD for each bin are plotted. This format is followed each time a quantity of interest is plotted as a function of $\tau_O - \tau_A$

## Discussion

We formalize SoA by drawing parallels from a Bayesian inference of the ventriloquism effect that estimates a common cause behind its multisensory integration. Understanding causality has been viewed to facilitate predictive, adaptable, and goal-directed actions[25–27]; hence, this may bring about SoA. Our Bayesian model integrates the action-outcome signals, compares them with the prior expectation, and infers the causality between them as well as the timing of these sensory signals. Our model could concisely reproduce the intentional binding experiments by Haggard et al.[3] and Wolpe et al.[22]. Whether intentional binding effects indeed follow Bayesian principles remained obscure.

Specifically, this was raised as an open question by Moore and Fletcher[14], pointing out only indirect empirical evidence existed in support of Bayesian cue integration, and Wolpe et al.[22] even posited that Bayesian cue integration does not explain outcome binding. Our model explains the temporal binding and repulsion phenomena as compromise between the noisy sensory observations and the prior belief of the action-outcome timing. Importantly, our Bayesian model predicts that the perceptual binding is generally trial-dependent and it must be correlated with the estimated causality $\hat{\xi}$ between the action and outcome. This prediction can be tested when the probability $P_c$ for detecting causality is not close to 0 or 1, by examining whether the

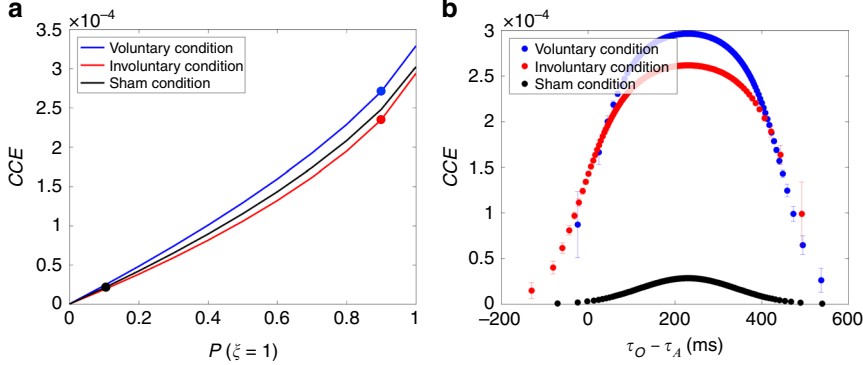

**Fig. 4** Bayesian model predictions of the confidence in causal estimate (*CCE*), which is our proposed measure for SoA. **a** Our Bayesian model predicts *CCE* to increase with a stronger causal prior. Furthermore, *CCE* differs for each condition even with equal prior strengths. This can be attributed to the difference in the amplitude of the jitter in the self-generated vs TMS-induced movement (muscle twitches) and audible clicks. **b** When plotted as functions of the trial-to-trial temporal disparity $\tau_O - \tau_A$, with the specific causal priors obtained for each condition, marked in **a**, *CCE* has a higher peak in the voluntary condition, but much lower values in the sham condition. Furthermore, *CCE* diminishes as the temporal disparity in sensory inputs moves further away from the prior mean $|\tau_O - \tau_A - \mu_{AO}|$. This falling of the *CCE* is faster when the causal prior is weaker and the uncertainty in the action input signal is higher

distribution of action-outcome intervals is bimodal and whether the intervals correlate with the reported causality between the action and outcome. We have therefore shown how Bayesian mechanism may underlie intentional binding. This is a significant contribution, as no previous Bayesian proposals accounted for experimental data on intentional binding and repulsion.

In addition, we theorize SoA as the *CCE*. *CCE* is high when the action-outcome timing is consistent with the causal prior, the causal prior is strong, and the action and outcome signals are reliable. This notion is consistent to what have been propounded as demonstrations of SoA: SoA arises from the causal relation between performed actions and their consequences[1,21,27,28], and from the integration of different agency cues whose individual influences are determined by their reliability[14,15,29–32]. Hence, we posit *CCE* to be a plausible measure of SoA. Here, Bayesian cue integration in terms of *CCE* is derived based on the computational principle of optimal inference in contrast to empirical observations that causality and reliability are involved. Further, *CCE* can explain outcome binding in terms of cue reliability that was previously considered non-Bayesian[22]. *CCE* is not an indicator of intention or a simple estimate of whether the action caused an outcome, but a new proposal of how SoA may emerge from the confidence in the estimate of the causality and timing (see discussion below on *CCE* against intention-based temporal binding).

Specifically, we postulate *CCE* fits the notion of a pre-reflective, implicit FoA. Synofzik et al.[1,30] provide a compelling account of such feeling: FoA is best accounted for by multimodal weighting and integration of different agency cues, and consists of an automatic registration of whether an action or sensory event is caused by the self or not. They posit FoA is nothing other than first-person in that the self is implied; hence, no external attribution (e.g., to TMS that caused the action) is possible. In the event that there is a feeling of exogenous causation, this will be overwritten by an explicit, interpretative judgment of agency (JoA) based on contextual beliefs or rationalizations. Similarly, the analytical expression of *CCE* shows that it is a multimodal weighting and integration process that lies at the center of obtaining a Bayesian causality inference. Furthermore, *CCE* itself does not attribute causality to any external agent, such as in the case of strong causal prior for TMS-induced movements. The judgement of the causality, $\hat{\xi}$, is then made based on the posterior ratio $r$ that compares *CCE* with the confidence in the acausal estimate. Perceptual timing in our model simply reflects the

sensory signals if the causality is not detected ($\hat{\xi} = 0$), whereas they are overwritten by the influence of the prior if the causality is detected ($\hat{\xi} = 1$). For example, in the involuntary condition of Haggard et al.[3], the estimated action and outcome timing by the model repulse reflecting the judgment of the causality. A compelling speculation in the paper by Haggard et al.[3] suggests this notion: the repulsion in the involuntary condition "reflects a mental operation to segregate, and thus to discriminate, pairs of events that cannot plausibly be linked by our own causal agency" (p. 384). We suggest such mental operation fits the notion of JoA, as quantified by the time shifts in Eq. (5) with the detected causality, and the peculiar feeling of causation by the involuntary movement to be FoA, quantified by *CCE*.

Following the above explanation, our theory therefore has a different take of the binding effect by Haggard et al.[3], which requires intentionality. Although intentional binding has been repeatedly observed in the context of voluntary action, it remains contentious in the literature whether it is indeed specific to voluntary action, or causality contributes to this effect[33]. Our model argues that the judgment of the causality is central to the perceived temporal action-outcome binding, consistent with current evidence that competes with the intentional account: the temporal binding is actually causal, not intentional[21,27,34]. For example, our model judges the causation of the tone even by the TMS-induced action in the involuntary condition. Hence, our Bayesian model predicts this unintended causality. Furthermore, our Bayesian model predicts that the action-outcome timing shifts toward the prior belief, $\hat{t}_O - \hat{t}_A \approx \mu_{AO}$, when the causality is perceived irrespective of the nature of the action, whether self-generated (i.e., the voluntary condition) or unintended (i.e., the involuntary condition). Interestingly, this temporal binding toward the same prior belief produces the compression and repulsion effects if the perceptual delay in the action timing ($d_A$) is small and large, respectively. What causes this difference in the perceptual delay? We found that unreliable senses (with large $\sigma_A$ or $\sigma_O$) tend to involve long perceptual delays (with large $d_A$ or $d_O$). Hence, the observed large perceptual delay in the TMS-induced action timing may be caused by the internal prediction error due to the absence of efference copy[35–37] and artificially perturbed neural activity. In this sense, intentionality is not strictly necessary for the sense of causality but influences the precision-dependent action-outcome timing shifts in our model. This is consistent with a recent empirical finding of intentional binding-like effects that emerged without intentional actions[33].

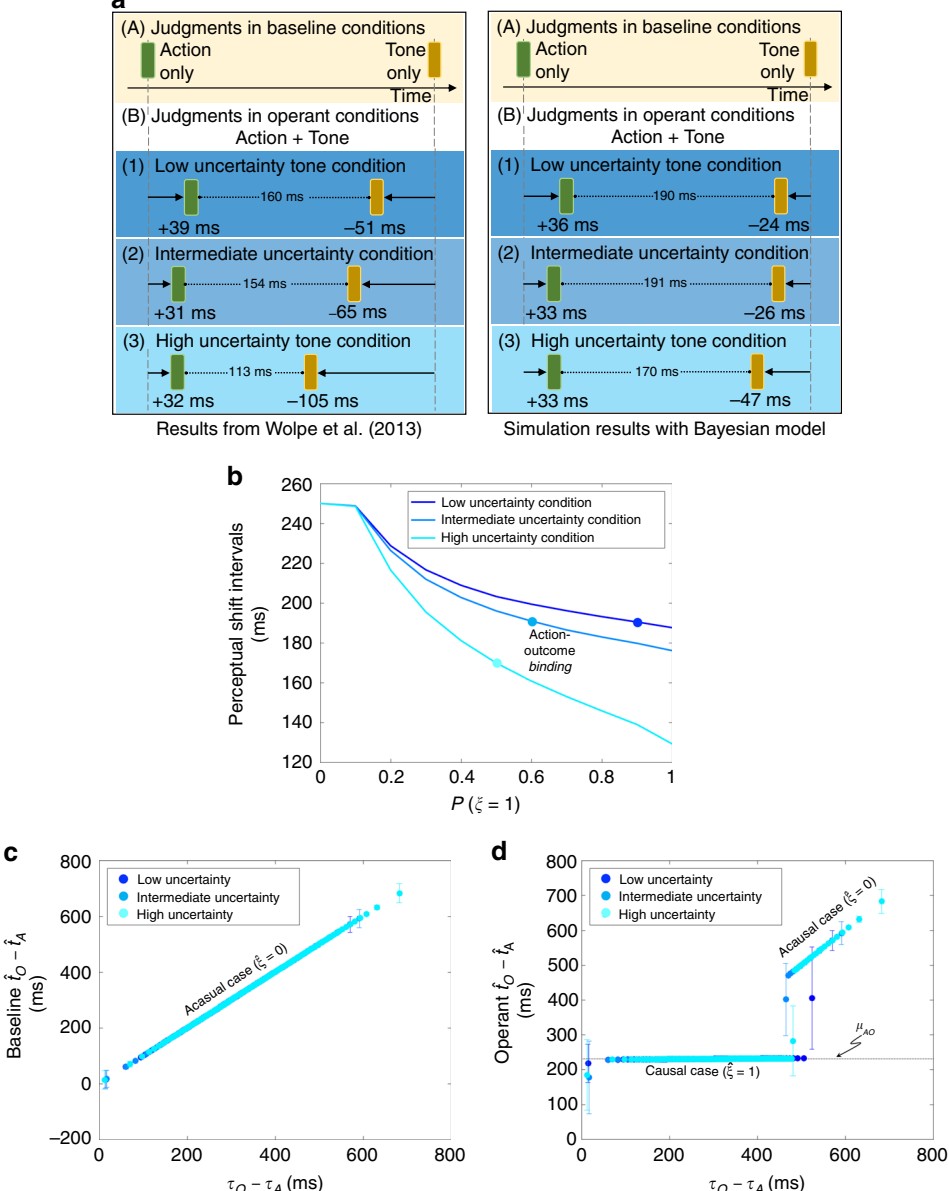

**Fig. 5** Qualitative replication of, as well as predictions related to, the results reported by Wolpe et al.[22]. **a** Qualitative replication of the experimental results (left panel) by our Bayesian model (right panel). **b** The action-outcome binding increases under heightened uncertainty. However, causality is less detected when the causal prior is lower, which decreases the action-outcome binding effect. The best estimates of the Bayesian model in **a** were obtained from different causal prior strengths, specifically $P(\hat{\xi} = 1)$ is 0.9, 0.6, and 0.5 (marked by the colored dots) for the low, intermediate, and high tone uncertainty conditions, respectively. **c**, **d** The causal prior strengths that correspond to each condition were used for the Bayesian estimate of the action-outcome timing interval $\hat{t}_O - \hat{t}_A$ in the baseline and operant conditions. The Bayesian estimate follows the sensory inputs in the baseline condition where all trials are acausal, but shifts towards the prior assumption, $\tau_O - \tau_A \approx \mu_{AO}$, when causality is detected. The temporal window of $\tau_O - \tau_A$ for detecting causality is wider when the outcome uncertainty is lower, which means more instances demonstrate binding

We predict that experimental manipulations that reduce $\sigma_A$ would increase perceived SoA even for unintended artificial actions. The prediction is therefore distinct from what was previously considered and can therefore serve as testable prediction for future experiments on causal agency.

Our theory also has a different take of the binding effect of Wolpe et al.[22]. Wolpe et al.[22] showed intentional binding as cue integration with uncertainty in outcome signals. They speculated that action and outcome bindings are driven by two distinct mechanisms: action binding is predicted by cue integration but outcome binding supports the predictive pre-activation hypothesis[38], i.e., the neural representation of the sensory outcome is activated prior to it. Hence, the outcome signals are perceived faster with less jitter than when it is not predicted to occur after the action. This could explain why the subjects' timing estimations are largely erroneous in the baseline condition and why the outcome binding is greater than the action binding. Our theory, although qualitative, explains both action and outcome bindings by a single Bayesian cue integration mechanism. Our model explains that the magnitudes of the action and outcome perceptual shifts, $\tau_O - \tau_A - \mu_{AO}$, are influenced primarily by the ambiguity of the outcome sensory signals, $(\sigma_A^2 + \sigma_O^2)/\sigma_{tot}^2$, and also in part by the strength of the causal prior that diminishes with outcome uncertainty.

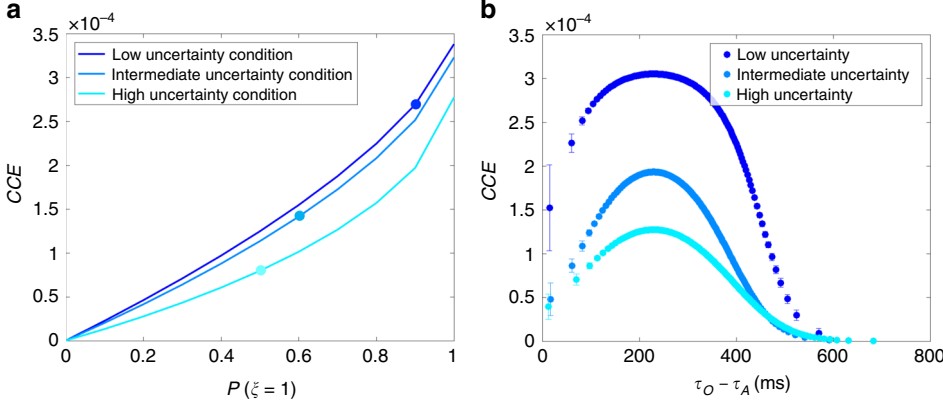

**Fig. 6** Bayesian model predictions of *CCE* as function of the causal prior and temporal disparity $\tau_O - \tau_A$. **a** The different effects of the causal prior on *CCE* across the three conditions is evident even with equal causal priors, which means that *CCE* depends on outcome uncertainty. **b** When plotted as functions of the temporal disparity $\tau_O - \tau_A$, given the condition-dependent causal priors (marked by the colored dots in **a**), *CCE* falls off with the disparity of sensory inputs from the prior mean, $|\tau_O - \tau_A - \mu_{AO}|$, faster when the outcome uncertainty is higher

The intentional binding paradigm has also been used to study pathological SoA[39–41]. Patients with schizophrenia tend to have much stronger temporal binding than healthy volunteers. Moreover, unlike healthy volunteers, their temporal binding of action timing does not depend on the probability of the outcome tone presentation[41]. These results are explained by our Bayesian model by assuming that schizophrenia patients cannot easily adapt their abnormally strong belief in causality (i.e., too large $P$ ($\xi = 1$)) and the uncertainty in the outcome (i.e., $\sigma_O$). Another important point is that, unlike healthy volunteers, patients with schizophrenia exhibit temporal binding of action timing that depends on the presence or absence of the outcome. It will be an interesting future study to model this result by explicitly incorporating the probabilistic occurrence of the outcome in our Bayesian model.

In summary, we posit that as the Bayesian cue integration is primarily precision-dependent so is our theory of SoA. Our model predicts and awaits confirmation that if the uncertainty of the sensory input signals could be maintained small, even unintended causal action may give rise to high *CCE* (hence, strong SoA)—hence, our notion of precision-dependent casual agency. We posited the precise estimation that gives rise to SoA encapsulates consistency in the perceived action-outcome effect, the prior belief of the causation of the outcome by the action, and the reliability of the perceived sensory signals. This theory may shed light on the mechanism of reduced SoA in psychosis, the understanding of the difference between FoA and JoA, and the design of prosthetic devices that heighten SoA. Furthermore, the challenge for future experiments that aim to link intentional binding to SoA is to demonstrate effects beyond what our model has already predicted: with the reliability of sensory inputs and strength of causal prior diminished, intentionality should be sufficient for strong intentional binding to emerge or not.

## Methods

**Analytical expressions for the Bayesian estimates**. The MAP estimate (Eq. (3)) of the Bayesian observer has a simple analytical expression. The MAP estimation is computed based on the posterior probability $P(t_A, t_O, \xi | \tau_A, \tau_O) = P(\tau_A, \tau_O, t_A, t_O, \xi) / P(\tau_A, \tau_O)$, where the peak location only depends on the joint distribution $P(\tau_A, \tau_O, t_A, t_O, \xi)$ in the numerator. The joint distribution is decomposed as $P(\tau_A, \tau_O, t_A, t_O, \xi) = P(\tau_A | t_A) P(\tau_O | t_O) P(t_A, t_O | \xi) P(\xi)$, where the conditional distributions for action and outcome are $P(\tau_A | t_A) = \exp\left[-(t_A - \tau_A)^2 / (2\sigma_A^2)\right] / \sqrt{2\pi\sigma_A^2}$ and $P(\tau_O | t_O) = \exp\left[-(t_O - \tau_O)^2 / (2\sigma_O^2)\right] / \sqrt{2\pi\sigma_O^2}$, respectively, and the prior

distribution is

$$P(t_A, t_O | \xi) = \begin{cases} \exp\left(-\dfrac{(t_O - t_A - \mu_{AO})^2}{2\sigma_{AO}^2}\right) / Z_1 & (\xi = 1) \\ 1/Z_0 & (\xi = 0) \end{cases}$$

with normalization constants $Z_1 \equiv \int_R \exp\left(-\dfrac{(t_O - t_A - \mu_{AO})^2}{2\sigma_{AO}^2}\right) dt_A dt_O \approx \sqrt{2\pi}\sigma_{AO} T$ and $Z_0 \equiv \int_R dt_A dt_O = T^2$.

The prior probability distribution $P(t_A, t_O | \xi)$ cannot be normalized unless a finite range of $(t_A, t_O)$ is defined. Therefore, we only consider it in the range $R = \{t_A, t_O | t_A \in (t_A^* - T/2, t_A^* + T/2), t_O \in (t_O^* - T/2, t_O^* + T/2)\}$ and assume that it is zero outside $R$, where again $t_A^* = 0$ ms and $t_O^* = 250$ ms are the true action and outcome timings, unknown to the observer, and $T = 250$ ms is a large enough but finite constant that specify the interval lengths in consideration. Hence, the prior probability distribution $P(t_A, t_O | \xi)$ must be normalized within $R$. Our results are robust to a shift in the center of $R$.

We separately compute the peak location $(\hat{t}_A, \hat{t}_O)$ for the causal case $\xi = 1$ and the acausal case $\xi = 0$ and, then, compare these two peaks. In the acausal case, because $P(\tau_A | t_A)$ and $P(\tau_O | t_O)$ take the maximum values at $t_A = \tau_A$ and $t_O = \tau_O$, respectively, the location of the acausal peak is $(\hat{t}_A, \hat{t}_O)|_{\xi=0} = (\tau_A, \tau_O)$ and the peak value is $\max_{t_A, t_O} P(\tau_A, \tau_O, t_A, t_O, \xi = 0) = \dfrac{P(\xi=0)}{2\pi\sigma_A\sigma_O Z_0}$. In the causal case, the peak of the joint distribution is found by minimizing a quadratic function. The peak location is given by $(\hat{t}_A, \hat{t}_O)|_{\xi=1} = \left(\tau_A + \dfrac{\sigma_A^2}{\sigma_{tot}^2}(\tau_O - \tau_A - \mu_{AO}), \tau_O - \dfrac{\sigma_O^2}{\sigma_{tot}^2}(\tau_O - \tau_A - \mu_{AO})\right)$, where $\sigma_{tot}^2 \equiv \sigma_A^2 + \sigma_O^2 + \sigma_{AO}^2$ is the total variance, and the peak value is computed as $\max_{t_A, t_O} P(\tau_A, \tau_O, t_A, t_O, \xi = 1) = \dfrac{P(\xi=1)}{2\pi\sigma_A\sigma_O Z_1} \exp\left(-\dfrac{(\tau_O - \tau_A - \mu_{AO})^2}{2\sigma_{tot}^2}\right)$. We define the log ratio of the posterior peaks for $\xi = 1$ and $\xi = 0$ by

$$r \equiv \frac{\max\limits_{t_A, t_O} P(t_A, t_O, \xi = 1 | \tau_A, \tau_O)}{\max\limits_{t_A, t_O} P(t_A, t_O, \xi = 0 | \tau_A, \tau_O)} = \exp\left(\theta - \frac{(\tau_O - \tau_A - \mu_{AO})^2}{2\sigma_{tot}^2}\right)$$

with $\theta \equiv \log\left[\dfrac{P(\xi=1)Z_0}{P(\xi=0)Z_1}\right]$. If $r > 1$, the MAP estimate is given by $(\hat{t}_A, \hat{t}_O)|_{\hat{\xi}=1}$ and $\hat{\xi} = 1$, which predicts perceptual shifts. If $r < 1$, the MAP estimate is given by $(\hat{t}_A, \hat{t}_O)|_{\hat{\xi}=0}$ and $\hat{\xi} = 0$, which predicts no perceptual shifts. The probability for detecting causality (i.e., $\hat{\xi} = 1$) is also easily computable, because $\tau_O - \tau_A - \mu_{AO}$ is distributed according to the Gaussian distribution $\mathcal{N}(m, \sigma_A^2 + \sigma_O^2)$ with $m \equiv t_O^* - t_A^* + d_O - d_A - \mu_{AO}$. Hence, the causality is detected if $|\tau_O - \tau_A - \mu_{AO}| < \sqrt{2\theta}\sigma_{tot}$ and this happens with probability

$$P_c = \frac{1}{2}\left[\text{erf}\left(\frac{\sqrt{2\theta}\sigma_{tot} - m}{\sqrt{2(\sigma_A^2 + \sigma_O^2)}}\right) + \text{erf}\left(\frac{\sqrt{2\theta}\sigma_{tot} + m}{\sqrt{2(\sigma_A^2 + \sigma_O^2)}}\right)\right].$$

Next, we evaluate the confidence in the causal MAP estimation $CCE \equiv \max\limits_{t_A, t_O} P(t_A, t_O, \xi = 1 | \tau_A, \tau_O)$, which comprises the numerator of the ratio $r$. To quantify this confidence, we need to first evaluate $P(\tau_A, \tau_O) = P(\tau_A, \tau_O, \xi = 1) +$

$P(\tau_A, \tau_O, \xi = 0)$ with

$$P(\tau_A, \tau_O, \xi = 1) = \iint_R P(t_A, t_O, \xi = 1, \tau_A, \tau_O) dt_A dt_O$$

$$= \frac{P(\xi=1)\sigma_{AO}}{Z_1 \sigma_{tot}} \exp\left(-\frac{(\tau_O - \tau_A - \mu_{AO})^2}{2\sigma_{tot}^2}\right)$$

and

$$P(\tau_A, \tau_O, \xi = 0) = \iint_R P(t_A, t_O, \xi = 0, \tau_A, \tau_O) dt_A dt_O$$

$$= \frac{P(\xi=0)}{Z_0}$$

Combining these expressions together, we obtain

$$CCE = \max_{t_A, t_O} P(\tau_A, \tau_O, t_A, t_O, \xi = 1)/P(\tau_A, \tau_O)$$

$$= \frac{\sigma_{tot}}{2\pi\sigma_A\sigma_O\sigma_{AO}} \text{Sigmoid}\left(\theta - \frac{(\tau_O - \tau_A - \mu_{AO})^2}{2\sigma_{tot}^2} + \log\frac{\sigma_{AO}}{\sigma_{tot}}\right)$$

where Sigmoid$(x) = 1/(1 + e^{-x})$ is the sigmoid function.

In this work, we focus on the timing to investigate the intentional binding effects but the mathematical elucidations above can permit other modalities (e.g., visual or haptic) and structural properties (e.g., inter alia, location, size, shape, and texture).

**Model fitting**. The simple analytical expression for the Bayesian timing estimate has an intuitive form and exposes all parameter dependencies explicitly. This allowed us to perform a theoretically guided parameter search to reproduce the experiments. We posit the perceptual delay $d$ and jitter of SD $\sigma$ due to sensory noise explain the reported means and SDs of the baseline event timing. Hence, we could immediately fix the values of parameters $d_A$, $\sigma_A$, $d_O$, and $\sigma_O$ (Table 1-Sets A and B). This leaves us with three free parameters, $\mu_{AO}$, $\sigma_{AO}$, and $P(\xi = 1)$, where fitting is not direct. Equation (5) shows that $\mu_{AO}$ alone can determine the qualitative difference between action-outcome binding ($\tau_O - \tau_A > \mu_{AO}$) and repulsion ($\tau_O - \tau_A < \mu_{AO}$). This immediately gives us a possible range of $\mu_{AO}$ that could account for both binding and repulsion, which is 182 ms < $\mu_{AO}$ < 259 ms, because $(t_O^* + d_O) - (t_A^* + d_A) - \mu_{AO}$ must be positive and negative in the voluntary condition and involuntary condition, respectively, from Eq. (5). We therefore tested $\mu_{AO}\epsilon[190, 200, \ldots, 240, 250]$ms with 10 ms increments. Our model also explains that both $\sigma_{AO}$ and $P(\xi = 1)$ can similarly influence the magnitude of binding and repulsion (c.f. Eq. (5) and formula for $P_c$). To obtain discernible perceptual shifts, $\sigma_{AO}$ should be small and $P(\xi = 1)$ should be large. As their effects are similar, we fixed $\sigma_{AO} = 10$ ms and we varied $P(\xi = 1)$ later on, and observed how different causal prior strengths influenced action-outcome binding and repulsion.

The principal measure of intentional binding is the mean perceptual shift of temporal awareness of action and sensory outcome. A perceptual shift is the change in the subjective estimation of action or outcome timing from the baseline to the operant condition. This can be computed as $E[\hat{t}_A] - E[\tau_A]$ and $E[\hat{t}_O] - E[\tau_O]$ (c.f. Eq. (5)) for action and outcome timings, respectively. A positive shift therefore informs the perception of timing shifted later in time and a negative shift informs the perception of timing shifted earlier in time. We could then compute for the model estimation error as absolute difference between our Bayesian model's estimates of the mean action and outcome perceptual shifts and the corresponding perceptual shifts reported in the experiments. We then selected the parameter values that best minimized the model estimation error.

**Simulation details**. Table 1 lists all the parameters of our Bayesian model. We performed different simulations to reproduce the action and outcome perceptual shifts reported by Haggard et al.[3] and Wolpe et al.[22], and to explain their underlying psychophysical mechanisms in Bayesian terms.

In the first simulation, our objective was to determine $\mu_{AO}$, to reproduce the perceptual shifts reported by Haggard et al[3]. We generated 35,000 instances of $\tau_A$ and $\tau_O$ pairs for each experimental condition using the baseline parameters in Table 1-Set A. Testing each value in the set of possible values for $\mu_{AO}$, and with $\sigma_{AO} = 10$ ms, we obtained the model estimation errors for the reported action and outcome perceptual shifts listed in Table 2-Set A. We took the average of the model estimation errors for the voluntary, involuntary, and sham conditions to obtain a single model estimation error. We looked at the model estimation errors for (a) action perceptual shifts only, (b) outcome perceptual shifts only, and (c) action-outcome perceptual shifts. Our results showed the best estimates of the model to be at $\mu_{AO} = 230$ ms. Furthermore, we observed our Bayesian model's estimates of the perceptual shift in action timing alone was sufficient to indicate the optimal parameters of the model.

Our objective in the second simulation was to obtain the specific strength of the causal prior that reproduces Haggard et al.'s results. With $\mu_{AO} = 230$ ms and $\sigma_{AO} = 10$ ms, we tested for $P(\xi = 1)$ in the range 0 to 1 with increments of 0.1. We used the same pairs of $\tau_A$ and $\tau_O$ from the first simulation, and we computed once again the model estimation errors for the empirical results listed in Table 2-Set A. We selected the $P(\xi = 1)$ that best minimized the model estimation errors for the voluntary, involuntary, and sham conditions, and fit the experimental data. Table 1-Set C includes the parameters that yielded the best model estimates.

| **Table 2 Reported perceptual shifts in action and outcome temporal awareness** | | |
| --- | --- | --- |
| **Judged event (operant condition)** | **Mean estimation error (in ms)** | **Mean perceptual shift (in ms)** |
| • Set A: Reported by Haggard et al.[3] | | |
| Voluntary action, then | 21 | 15 |
| Tone | −31 | −46 |
| Involuntary action, then | 56 | −27 |
| Tone | 46 | 31 |
| Sham TMS, then | 25 | −7 |
| Tone | 7 | −8 |
| • Set B: Reported by Wolpe et al.[22] | | |
| Action, then | 31 | 39 |
| Low uncertainty tone | −16 | −51 |
| Action, then | 23 | 31 |
| Intermediate uncertainty tone | −19 | −65 |
| Action, then | 24 | 32 |
| High uncertainty tone | −10 | −105 |

Figure 1 shows the action and outcome perceptual shifts, as well as the intervals between perceptual shifts, which were obtained by our Bayesian model using these parameters.

In the third simulation, we aimed to reproduce the perceptual shifts reported by Wolpe et al.[22], listed in Table 2-Set B. We generated another set of 35,000 $\tau_A$ and $\tau_O$ pairs using this time the baseline parameters listed in Table 1-Set B. We performed simulations with $\mu_{AO} = 230$ ms, $\sigma_{AO} = 10$ ms, and $P(\xi = 1)$ in the range 0 to 1 with increments of 0.1. We did not perform additional simulations to redetermine $\mu_{AO}$, as our aim is to reproduce qualitatively all the experiments with the same $\mu_{AO}$ and $\sigma_{AO}$ as possible in order to have simple yet consistent explanations by our Bayesian model. Although we did not modify here $\mu_{AO}$ and $\sigma_{AO}$, our analyses and results can show their effects can be predicted and explained by our model. The model estimation errors once again indicate the estimates of action perceptual shifts led to the best estimates of the model. We list under Table 1-Set D the $P(\xi = 1)$ that yielded the best estimates of the model for the low, intermediate, and high uncertainty tone conditions. Figure 5a shows the action and outcome perceptual shifts, and intervals between shifts, predicted by our Bayesian model for this experimental setup.

In the fourth simulation, our objective was to determine the influence of the causal prior and the temporal difference $\tau_O - \tau_A$ (that varies in every trial) on the various predictions of our Bayesian model for Haggard et al.'s experimental setup. We used the model parameters and $\tau_A$ and $\tau_O$ pairs from the first and second simulations. We obtained our Bayesian model's predictions of the intervals between action and outcome perceptual shifts, binding and repulsion effects, action-outcome timing interval, $\hat{t}_O - \hat{t}_A$, in the baseline and operant conditions, and $CCE$. The results are shown in Figs. 2–4.

Our objective and target results in the final simulation were the same as the fourth simulation, but we used the model parameters and $\tau_A$ and $\tau_O$ pairs from the third simulation to account for the experimental setup of Wolpe et al.[22]. The resulting plots are shown in Figs. 5 and 6.

**Reporting summary**. Further information on research design is available in the Nature Research Reporting Summary linked to this article.

## Data availability
All relevant data are within the manuscript, which can be immediately generated using the supplementary MATLAB source codes.

## Code availability
The MATLAB source codes that were used to generate the simulated datasets and analyze the simulation results are appended as Supplementary Source Codes.

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

## Acknowledgements

This study was supported by Brain/MINDS from AMED under Grant Number JP19dm020700 and JSPS KAKENHI Grant Number JP18H05432.

## Author contributions

R.L. and T.T. planned the project, conceived the Bayesian model, performed the simulations, analyzed the data, and wrote the manuscript.

## Additional information

**Competing interests:** The authors declare no competing interests.

