## [Transparent Peer Review File · Nature Communications]

Reviewers' comments:

Reviewer #1 (Remarks to the Author):

The submitted paper by Legaspi and Toyozumi presents an intriguing idea: what if the effect of the sense of agency, e.g. differences in integration of signals if initiated by self or others, is all due to differences in timing and variability of sensory signals. They propose a Bayesian causal inference model for psychophysics, and show that it can explain the main findings from two previous studies.

Using Bayesian causal inference to explain the sense of agency is a good idea (and seems obvious in hindsight!), but I suspect there will be effects in the literature that can not be explained. Having the model be fully agnostic between self-intended and unintended actions may need to be addressed in the future.

However, the modelling is appropriate, and the paper is interesting and worth publishing.

Major points:

The model is based on a paper by Sato et al that studied the ventriloquist illusion. While mathematically that is true, the spiritual predecessor is surely the Kording et al. 2007 paper, which describes a near identical model in terms of causal inference (the Sato paper never discusses causality). The Kording et al 2007 paper should at least be referenced when discussing causality. I would also suggest referencing background material on modelling causality, e.g. Pearl 2000.

The section on model fitting is a little sparse, just mentioning that it is done using absolute error. Was there no quantitative fitting done (e.g. gradient descent)? Or was it just a matter of trying different values until something was close?

The authors assume that the subject will choose that most likely causal structure. Another option could be model averaging (see again Kording et al.). Could the authors comment on how that would change the results?

Likewise the calculation of the confidence in causal estimate (CCE) could as well be done in a fully Bayesian way by marginalising out t_A and t_O , instead of taking the max.

Minor comments:

Page 7, eq 2: specify μ_{A0} here, not just in Methods

page 9: Reference is made to a non-existing Appendix

page 10, line 180, awkward sentence

page 10, line 186: " perceptual binding in the action-outcome interval ", what is quantitatively meant by perceptual binding? I do not see the link with eq 5

page 13: When describing the tone uncertainty in Wolpe et al. make it clear why the uncertainty level doesn't just affect the temporal variance, as one might assume it would.

figure 1, line 601: awkward sentence

page 22: please add units (e.g. ms) when describing values

Ulrik Beierholm
Durham University

Reviewer #2 (Remarks to the Author):

This paper formalises a Bayesian model of sense of agency.

There are two main reactions.

1. The idea of Bayesian processes in sense of association is not new: it is explicit in papers by Wolpe et al, and by Moore et al. The current paper has the merit of formalising the presentation, but the ideas appear to be quite familiar. There is no new data. Then, the question becomes how much of a contribution it is to add a formalisation. I note that many papers in this area involve modelling and data - formalisation alone is a contribution, but perhaps not a sufficient one. In addition, how much explanatory value does this formalisation add? For example, a CCE is posited to account - but this seems in the end to be just giving a name to the intuition that actions cause their outcomes. Naming the prior does not explain how the mechanism works.

2. The MS appears to make a mistake in thinking that intentional binding is exclusive, i.e., that perceptual compression will occur ONLY in the case of intentional action. This perhaps shows a naivety about the wider time perception literature. There are MANY factors that influence the perceived time of an event. One would predict a temporal compression of the A-B interval whenever there is an association between A and B: an idea that goes back to Pavlov or even Locke, and is certainly more general than intentional binding. Thus, binding is commonplace. The point about intentional binding is that binding is stronger when an action is made intentionally than when it is not. This is entirely compatible with the idea that unintentional actions, causal events unrelated to action, and other scenarios, all show some degree of binding, and indeed a number of studies have shown effects of this kind. Good scientific arguments are based not on showing whether there is binding in condition X or not, but in designing two conditions X and Y that differ in one and only one respect, and then investigating whether they show different levels of binding. If they do, then the factor in which those conditions differ would appear to be relevant to time perception (and potentially to SoA or whatever time perception is being used as a proxy for).

Response to Reviews

We thank the reviewers for providing valuable comments that can only help improve our work and manuscript. We have provided below our point-by-point response to all their comments.

Reviewer #1 (Remarks to the Author):

The submitted paper by Legaspi and Toyoizumi presents an intriguing idea: what if the effect of the sense of agency, e.g. differences in integration of signals if initiated by self or others, is all due to differences in timing and variability of sensory signals. They propose a Bayesian causal inference model for psychophysics, and show that it can explain the main findings from two previous studies.

Using Bayesian causal inference to explain the sense of agency is a good idea (and seems obvious in hindsight!), but I suspect there will be effects in the literature that cannot be explained. Having the model be fully agnostic between self-intended and unintended actions may need to be addressed in the future. However, the modelling is appropriate, and the paper is interesting and worth publishing.

We thank the reviewer for deeming our work worthy of publication.

Major points:

The model is based on a paper by Sato et al that studied the ventriloquist illusion. While mathematically that is true, the spiritual predecessor is surely the Kording et al. 2007 paper, which describes a near identical model in terms of causal inference (the Sato paper never discusses causality). The Kording et al 2007 paper should at least be referenced when discussing causality. I would also suggest referencing background material on modelling causality, e.g. Pearl 2000.

Thank you for pointing this out. Indeed, our formulations are true to the original form used by Sato, Toyoizumi & Aihara (2007) for the Bayesian inference of the ventriloquism effect. Their prior indicated whether there is a common *source* of the audiovisual stimuli, which is akin to a common *cause* of the multisensory binding. Hence, it was intuitive for us to transition their prior from denoting a common source to one of causality. The significance, however, is that we drew the same prior that inferred the ventriloquism effect to infer intentional binding. However, it is also true that causality was pointed out explicitly by Kording et al. in 2007. Hence, we referred to their paper in the revised manuscript. We also added pertinent background material on causality modeling.

The section on model fitting is a little sparse, just mentioning that it is done using absolute error. Was there no quantitative fitting done (e.g. gradient descent)? Or was it just a matter of trying different values until something was close?

The strong point of our study is that the analytical expression for the inferred action and outcome timing is available in a very intuitive form, where all the parameter dependency is explicitly visible. Hence, we could perform theoretically guided parameter search. We have three free parameters μ_{AO} , σ_{AO} , and $P(\xi = 1)$ to reproduce the experiments. Equation (5) tells that only μ_{AO} can determine the qualitative difference between action-outcome binding

and repulsion. This immediately gives a possible range of μ_{AO} that reproduces the experimentally observed binding or repulsion results. The roles of other two parameters σ_{AO} and $P(\xi = 1)$ are largely overlapping as summarized by Equation (5) and the formula of P_C on p. 21. Hence, we fixed σ_{AO} and plotted the quantitative effect of $P(\xi = 1)$ in our figures. This is explained in the text in the following paragraph (on p. 9). We clarified further in the revised manuscript (on p. 22) how we exploited the analytical results to set the parameters.

Fitting of d_A , d_O , σ_A and σ_O is straightforward, they are suggested by the means and standard deviations of the reported subjects' baseline estimation errors (Table 1-Sets A and B). After fixing these parameters, the model is left with three free parameters, μ_{AO} , σ_{AO} , and $P(\xi = 1)$. As described in equation (5), μ_{AO} has an important role in determining if binding or repulsion happens in each experimental condition. A fixed value of $\mu_{AO} = 230$ ms successfully accounts for this qualitative behavior in all the 6 experimental conditions (3 from Haggard et al. and 3 from Wolpe et al.) that we study. The analytical expressions in Methods suggest that σ_{AO} and $P(\xi = 1)$ have a largely overlapping role in detecting causality. Causality is more likely detected if σ_{AO} is small or $P(\xi = 1)$ is large, although the exact mechanisms are slightly different. At least one of these two parameters need to be adjusted according to the conditions to account for the experimental observations. For simplicity, we fix $\sigma_{AO} = 10$ ms to be a small enough constant to permit noticeable perceptual shift and adjust $P(\xi = 1)$ (see Table 1 for the parameter values in 6 experimental conditions) to account for two observations in each condition, namely, the perceptual shifts in the action timing and the outcome timing.

The authors assume that the subject will choose that most likely causal structure. Another option could be model averaging (see again Kording et al.). Could the authors comment on how that would change the results? Likewise the calculation of the confidence in causal estimate (CCE) could as well be done in a fully Bayesian way by marginalising out t_A and t_O , instead of taking the max.

This is an interesting point. Model averaging was used both in Sato et al. and Körding et al. to account for the ventriloquism effect but, here, we chose not to take the averaging. Namely, we formulated the Bayesian observer to simultaneously infer the timing and causality. The reason is that subjects were explicitly asked to report the timing in the experiments we modeled. The reason for simultaneously inferring the causality is that we experience sense of agency, either implicitly or explicitly, even when not asked by an experimenter. This spontaneous sense of agency has implications for motivation, mental diseases, and social interaction. This formulation also has a technical benefit that the MAP estimator has simple analytical form, so that it is easy to analyze. Regarding the second issue, we can easily compute a variant of CCE after marginalizing t_A and t_O – all necessary ingredients to analytically evaluate this quantity are in fact already presented in Methods. However, we think that the current definition of CCE without marginalization is likely more relevant for the sense of agency as it is not only detecting causality but also is sensitive to the accuracy of the time estimation. This discussion for precision-dependent causal agency is elaborated in the revised manuscript (on p.9).

Minor comments:

Page 7, eq 2: specify μ_{AO} here, not just in Methods

We specified on p. 10 that $\mu_{AO} = 230$ ms while describing how we reached the parameter values used in our simulations. We discussed on p.7 only the theoretical description of μ_{AO} (and of the other model parameters).

page 9: Reference is made to a non-existing Appendix

The sentence was revised to correctly refer to Methods (on p.9, line 157).

page 10, line 180, awkward sentence

The sentence was revised in the main text (on p. 10, line 189), as follows:
Consistent to their findings, our Bayesian observer inferred the perceived action and outcome timings to shift towards each other in the voluntary condition resulting to compressed temporal intervals between the action and outcome perceptual shifts.

page 10, line 186: ‘perceptual binding in the action-outcome interval’, what is quantitatively meant by perceptual binding? I do not see the link with eq 5

Thank you for pointing this out. We are referring to the magnitude of temporal binding or repulsion, which is the difference in the perception of action and outcome timings in the baseline and operant conditions. From equation (5), this is given by $(\tau_O - \tau_A) - (\hat{t}_O - \hat{t}_A)$, which is $(\tau_O - \tau_A - \mu_{AO})(\sigma_A^2 + \sigma_O^2)/\sigma_{tot}^2$ in the causal case ($\hat{\xi} = 1$) and 0 in the acausal case ($\hat{\xi} = 0$). We incorporated this clarification in the revised manuscript (on p. 10, line 196).

page 13: When describing the tone uncertainty in Wolpe et al. make it clear why the uncertainty level doesn’t just affect the temporal variance, as one might assume it would.

We explained that tone uncertainty in Wolpe et al. accounted for the reduced causal prior if subjects were uncertain about the occurrence of the tone, and hence missed the causation. However, as the reviewer suggested, tone uncertainty could instead increase temporal jitter in perception through σ_{AO} . In fact, this has a similar effect to a weaker causal prior because $P(\xi = 1)$ and σ_{AO} have similar effect on the temporal binding as we discussed (on p. 10, line 179). Therefore, we only changed $P(\xi = 1)$ in this study and clarified in the text that σ_{AO} can be changed. We added “(Alternatively, we could increase σ_{AO} , which produces similar results; see above discussion on model fitting.)” in the text (on p. 13, line 271).

figure 1, line 601: awkward sentence

The sentence in the figure caption was revised (on p. 30, line 654), as follows:
Each subject’s mean judgment error in the single-event baseline condition was subtracted from the mean judgment error for the corresponding event in the operant condition. This resulted to the values underneath the blocks that indicate the magnitude and direction to which the temporal perceptions shifted.

page 22: please add units (e.g. ms) when describing values

The units were incorporated accordingly.

Reviewer #2 (Remarks to the Author):

This paper formalises a Bayesian model of sense of agency.

There are two main reactions.

1. The idea of Bayesian processes in sense of association is not new: it is explicit in papers by Wolpe et al, and by Moore et al. The current paper has the merit of formalising the presentation, but the ideas appear to be quite familiar. There is no new data. Then, the question becomes how much of a contribution it is to add a formalisation. I note that many papers in this area involve modelling and data - formalisation alone is a contribution, but perhaps not a sufficient one. In addition, how much explanatory value does this formalisation add? For example, a CCE is posited to account - but this seems in the end to be just giving a name to the intuition that actions cause their outcomes. Naming the prior does not explain how the mechanism works.

While we agree that Bayesian integration was proposed as a general principle behind sense of agency, it was unknown if intentional binding experiments (attraction or repulsion of action-outcome timings) are consistent with the Bayesian principle, and if so, how. Specifically, this was raised as an open question in (Moore and Fletcher, 2012) and there is even a discussion that Bayesian cue integration does not explain the outcome binding effect (Wolpe et al., 2013). Hence, what we provide is not a formalism but a new theory demonstrating that Bayesian cue integration explains the intentional binding experiments and providing specifically how. We believe this is a significant contribution because no previous Bayesian proposals accounted for experimental data about intentional binding from qualitatively distinct conditions. Our model can explain both binding and repulsion effects (Haggard et al., 2002) and cue-uncertainty effect (Wolpe et al., 2013) by the same Bayesian causal prior. We propose *CCE* as an indicator of sense of agency and this is not just giving a name to an existing intuition – this is an alternative hypothesis to explaining the action-outcome binding effect in terms of intention. In terms of *CCE*, the Bayesian cue integration can also explain action-outcome binding by cue-reliability (Wolpe et al., 2013), which was previously considered non-Bayesian. Our proposal is neither a formality issue nor an interpretation matter. As a consequence, our theory gives specific predictions that are distinct from what was previously considered: (1) experimental manipulations that reduces the unreliability of sensory inputs would increase sense of agency even for unintended actions, and (2) intentional binding happens on a per-trial basis, yielding a bimodal distribution of the perceived action-outcome interval, and these two distinct peaks reflect presence and absence of sense of agency. These can therefore serve as testable predictions for future experiments on sense of agency. We incorporated revisions in the manuscript to clarify the novelty and advance of our work.

2. The MS appears to make a mistake in thinking that intentional binding is exclusive, i.e., that perceptual compression will occur ONLY in the case of intentional action. This perhaps shows a naivety about the wider time perception literature. There are MANY factors that influence the perceived time of an event. One would predict a temporal compression of the A-B interval whenever there is an association between A and B: an idea that goes back to Pavlov or even Locke, and is certainly more general than intentional binding. Thus, binding is commonplace. The point about intentional binding is that binding is stronger when an action is made intentionally than when it is not. This is entirely compatible with the idea that unintentional actions, causal events unrelated to

action, and other scenarios, all show some degree of binding, and indeed a number of studies have shown effects of this kind. Good scientific arguments are based not on showing whether there is binding in condition X or not, but in designing two conditions X and Y that differ in one and only one respect, and then investigating whether they show different levels of binding. If they do, then the factor in which those conditions differ would appear to be relevant to time perception (and potentially to SoA or whatever time perception is being used as a proxy for).

We thank the reviewer for pointing out a possible source of confusion. We do not think temporal binding arises solely from intentional actions since we are cognizant of the competing accounts in the literature about temporal binding, as evidenced for example in the following paragraphs:

Following the above explanation, our theory therefore has a different take of Haggard et al.'s binding effect that requires intentionality. CCE argues that the judgment of the causality is central to the perceived temporal binding, consistent with current evidence that competes with the intentional account: the temporal binding is actually causal, not intentional²².

- on p. 16, lines 338 and following of the original manuscript

Indeed, there are several factors that can lead to temporal binding. However, our point, and also our testable theory, is that Haggard's *intentional binding* per se, which is believed to arise solely from intentional actions (as opposed to involuntary ones), is simply explained by our model as a strong case of *precision*-dependent multisensory *causal* integration – hence, not by causation alone but also by the reliability of the sensory inputs. Moreover, even the involuntary condition that led to perceived temporal *repulsion* is explained by this same mechanism. Hence, two different conditions, voluntary and involuntary, that differ in the presence of intention (or lack thereof), and showed opposite perceptions of binding, are actually explained by the single mechanism of our Bayesian model. This contribution is not at all trivial, in fact, a very recent paper by Suzuki et al. (2019) (has been included in References) posits “Intentional binding without intentional action” (also the paper's actual title), arguing that it remains contentious in the literature whether intentional binding indicates intentionality or causality. While this paper resolved the issue empirically, our theory has predicted this phenomenon based on Bayesian principles. The challenge therefore for future experiments that aim to connect intentional binding to sense of agency is to provide testimony for effects beyond what our model already predicts: with reliability of sensory inputs and strength of causal prior diminished, intentionality should be sufficient for strong intentional binding to arise or not.

Taken into account both major comments of the reviewer, we acknowledge that many things can generally influence perceived time of an event and many formalisms can be added within a general Bayesian framework. In this sense, everything can be included in the Bayesian framework. However, a very general model has little explanatory and predictive power, and hence is not falsifiable. What we intended in this study is to propose that the causal prior as a parsimonious key element to explain how it reproduces intentional binding effects observed in different experimental conditions and to propose confidence in causal estimate as an indicator of sense of agency, which behaves distinct from previous proposals.

REVIEWERS' COMMENTS:

Reviewer #1 (Remarks to the Author):

All my comments have been addressed.

Note that 'resulting/resulted to' on pages 10 and 30 should be changed to 'resulting/resulted in'

Ulrik Beierholm

Reviewer #2 (Remarks to the Author):

The authors have made a few changes, but they are relatively superficial, and do not really address my concerns.

The critical point they make is that binding should occur when there is high confidence of an action causing an outcome. That is a useful prediction, but unfortunately the authors do not actually test it! They cite ref. 32, but (a) actually they might also cite the paper that reported this effect first, which is <https://www.ncbi.nlm.nih.gov/pubmed/28501698>

(b) they accept the findings of ref. 32 somewhat uncritically. A very plausible interpretation of ref. 32 is that the virtual reality setup used results in a *simulated* intentional action, and that this simulated intention may be sufficient for intentional binding. In that case, they cannot claim that intention is not necessary. In fact, the wealth of literature on 'mirror neuron' type effects in the literature would suggest that is exactly what happens in the situation of ref 32. Thus, I feel it is naive to use ref. 32 as a validation of the authors' prediction.

The authors do not adequately consider a number of studies in which an active keypress-then-tone produces more binding than a passive keypress-then-tone using the same response key. In such cases, everything is matched except for the efferent motor command. The authors' account would presumably argue that one has high confidence in keypress-causes-tone in the former case, but lower confidence in the latter case. It seems to me unlikely, but the authors should in any case discuss some evidence to convince the reader about this point.

That is, the authors seem to want to attribute the effect to the CCE, not to the efferent motor command. The authors' argument would need some *independent* evidence that confidence in keypress-causes-tone really does differ between active and passive conditions. Without such evidence, I feel that the authors' account is just too speculative for the journal.

Response to REVIEWERS' COMMENTS

Reviewer #1 (Remarks to the Author):

All my comments have been addressed.

Note that 'resulting/resulted to' on pages 10 and 30 should be changed to 'resulting/resulted in'

Ulrik Beierholm

Thank you for all the helpful comments. We corrected the typo on pp. 10 and 31.

Reviewer #2 (Remarks to the Author):

The authors have made a few changes, but they are relatively superficial, and do not really address my concerns.

The critical point they make is that binding should occur when there is high confidence of an action causing an outcome. That is a useful prediction, but unfortunately the authors do not actually test it! They cite ref. 32, but (a) actually they might also cite the paper that reported this effect first, which is <https://www.ncbi.nlm.nih.gov/pubmed/28501698> (b) they accept the findings of ref. 32 somewhat uncritically. A very plausible interpretation of ref. 32 is that the virtual reality setup used results in a *simulated* intentional action, and that this simulated intention may be sufficient for intentional binding. In that case, they cannot claim that intention is not necessary. In fact, the wealth of literature on 'mirror neuron' type effects in the literature would suggest that is exactly what happens in the situation of ref 32. Thus, I feel it is naive to use ref. 32 as a validation of the authors' prediction.

We explained in the manuscript that recent studies have already shown the perception of causality sufficient to elicit binding effect (e.g., Refs. [20,26,30,33]). Regarding the intentionality requirement, the reviewer proposed a counter hypothesis to Suzuki et al. (Ref. [32]), i.e., intentionality was attributed to the virtual hand, which resulted to a perceived binding effect by the subject. We are therefore correct to use Ref. [32] to drive the point that intentionality versus causality remains a contentious point and both are consistent with the existing experimental data. It is indeed interesting to test these hypotheses in a future experiment in which a subject can rate the degree of intentionality of the virtual hand that is empathetically perceived and study how it is modulated, for example, by the smoothness of the reaching trajectory.

The authors do not adequately consider a number of studies in which an active keypress-then-tone produces more binding than a passive keypress-then-tone using the same response key. In such cases, everything is matched except for the efferent motor command. The authors' account would presumably argue that one has high confidence in keypress-causes-tone in the former case, but lower confidence in the latter case. It seems to me unlikely, but the authors should in any case discuss some evidence to convince the reader about this point.

That is, the authors seem to want to attribute the effect to the CCE, not to the efferent motor command. The authors' argument would need some *independent* evidence that confidence in keypress-causes-tone really does differ between active and passive conditions. Without such evidence, I feel that the authors' account is just too speculative for the journal.

We are neither (1) claiming the strength of temporal binding to lie on *CCE* nor (2) neglecting the importance of the efference copy. We apologize for the confusion on (1). We clarified in the revised main text (on p. 17, quoted below) that the greater temporal binding for voluntary action than involuntary action does not directly stem from higher *CCE* but from shorter delay in action timing estimation (d_A). Importantly, in previously published experimental results, action timing delay is correlated with the imprecision (σ_A) of estimated action timing – we postulate this may be a general property. Our model suggests that, while sense of causality is necessary for an action-outcome timing shift, the degree of compression is dependent on d_A . Regarding (2), the role of efference copy that the reviewer suggests is perfectly consistent with our proposal. We argue that involuntary action lacking efference copy may cause a large prediction error in the comparator model that compares reafference and efference copy and, hence, can produce experimentally observed large σ_A .

“Furthermore, our Bayesian model predicts that the action-outcome timing shifts toward the prior belief, $\hat{t}_O - \hat{t}_A \approx \mu_{AO}$, when the causality is perceived irrespective of the nature of the action, whether self-generated (i.e., the voluntary condition) or unintended (i.e., the involuntary condition). Interestingly, this temporal binding toward the same prior belief produces the compression and repulsion effects if the perceptual delay in the action timing (d_A) is small and large, respectively. What causes this difference in the perceptual delay? We found that unreliable senses (with large σ_A or σ_O) tend to involve long perceptual delays (with large d_A or d_O). Hence, the observed large perceptual delay in the TMS-induced action timing may be caused by the internal prediction error due to the absence of efference copy³⁵⁻³⁷ and artificially perturbed neural activity. In this sense, intentionality is not strictly necessary for the sense of causality but influences the precision-dependent action-outcome timing shifts in our model.”